# The diurnal cycle and temperature dependence of crystal shapes in ice clouds from satellite lidar polarized measurements

Vincent Noel<sup>1</sup>, Hélène Chepfer<sup>2</sup>, Christelle Barthe<sup>1</sup>, John Yorks<sup>3</sup>

<sup>1</sup>LAERO, Univ Toulouse, CNRS, IRD, Toulouse, France

5 <sup>2</sup>LMD/IPSL, Sorbonne Université, Ecole Polytechnique, Institut Polytechnique de Pairs, ENS, PSL Université, CNRS, Paris, France

<sup>3</sup>NASA Goddard Space Flight Center, Greenbelt, Maryland, USA

Correspondence to: Vincent Noel (vincent.noel@cnrs.fr)

Abstract. The shape of crystals in ice clouds influences many aspects of the cloud lifecycle and radiative impact, yet they are extremely variable and hard to categorize. In this paper, we apply a recent crystal shape classification methodology to 33 months of spaceborne lidar measurements. We take advantage of their non-sun-synchronous nature to document the diurnal variability of ice crystal shapes. We find that that mid-level clouds are dominated by 3D bullets and 2D columns, with more 3D bullets at higher latitudes, in agreement with previous results. Shape dependence on latitude is generally symmetric around the equator. We document the repartition of shapes with temperatures, and show that the proportion of complex shapes (Droxtals and Voronois) increases at colder temperatures, becoming dominant below -60°C. Finally, we document the diurnal cycle of the repartition of shapes according to temperature and latitude. We find that 2D plates and columns appear preferentially during daytime, while complex shapes are more likely to appear during nighttime. 3D bullets follow a unique behavior, shifting from a daytime maximum at coldest temperatures to a nighttime maximum at warmer temperatures. The amplitude of diurnal cycles generally strengthens at colder temperatures. These results provide new constraints for the representation of ice clouds in atmospheric and climate models.

# 1 Introduction

Clouds in the upper part of the troposphere are largely made of ice particles, which can adopt an infinite variety of size and shape combinations (van Diedenhoven, 2018; Cairo et al., 2023). The microphysical properties of a given cloud are closely linked to its capacity for water vapor uptake and formation speed, its radiative impact and optical signature, and eventually its sedimentation and dissipation process (Gettelman et al., 2024). Particle shapes thus influence the life cycle of ice clouds. The difficulty in creating appropriate categories of crystal shapes in ice clouds makes their representation in atmospheric regional models complex, leading to significant uncertainties and errors when simulating the lifecycle of ice clouds and estimating their impact on other atmospheric processes (Taufour et al., 2024). If we want to properly account for the role of ice clouds in short-term atmospheric processes key to weather and extreme events prediction, it is essential to better

https://doi.org/10.5194/egusphere-2025-5018 Preprint. Discussion started: 14 November 2025

understand how ice crystal properties are spatially distributed throughout the globe, and which external atmospheric parameters influence them (Krämer et al., 2016). One area that has received little attention so far is whether the microphysical properties of ice clouds follow a diurnal cycle.

Recent advances in the simulation of the optical signature of ice particles have enabled the development of a particle type classification methodology, based on polarized lidar measurements (Okamoto et al., 2019). This methodology was applied successfully to several years of observations from the Cloud-Aerosol Lidar and Infrared Pathfinder Satellite Observations (CALIPSO) spaceborne lidar (Winker et al., 2009), leading to maps of the relative concentrations in ice clouds of specific particle types (Sato and Okamoto, 2023). Here we apply this methodology on ice clouds detected in measurements from the CATS lidar (Cloud-Aerosol Transport System, Yorks et al. 2016), to document how the repartition of particle types in ice clouds vary diurnally with the local time of observation, latitude, and temperature. Our objective is to better understand whether the repartition of ice crystal shapes in ice clouds follows a diurnal cycle.

After introducing the CATS dataset (Sect. 2.1) and the cloud detection and particle type classification methods (Sect. 2.2), we compare in Sect. 3.1 our results with those obtained from CALIPSO data by Sato and Okamoto (2023). We then describe the temperature dependence of the particle type partitioning (sect. 3.2) and its diurnal cycle (Sect. 3.3). We conclude in Sect.

45 4.

# 2. Data and Methodology

## 2.1 Backscatter data from the CATS spaceborne lidar

The CATS spaceborne lidar was operated from the International Space Station (ISS) between February 2015 and October 2017, leading to ~33 months of measurements of vertical profiles of attenuated backscatter. Over the complete period, most months were sampled 3 times, while months between November and February were sampled only twice: when considering the entire CATS dataset, results might be slightly dominated by the March-October period. Although initially equipped with 355 nm (UV), 532 nm (visible) and 1064 nm (IR) channels, due to technical difficulties the near-totality of CATS measurements (31 months) were performed at 1064 nm only. Laser performance and data acquisition were optimized for that wavelength, leading to high-quality measurements of attenuated backscatter coefficients (Pauly et al. 2019). Unlike CALIPSO, which as part of the sun-synchronous A-Train always took measurements at the same local time (01:30 AM/PM until 2018), ISS-based CATS samples were made at variable local times, giving access to the monthly or seasonal mean diurnal variations of observed cloud properties (Noel et al., 2018; Chepfer et al. 2019). CATS detections of ice clouds made at 01:30 AM/PM are very consistent with those based on CALIPSO (Sellitto et al., 2020). The ISS orbital inclination constrains CATS measurements zonally between 55°S and 55°N.

The results presented in the rest of this article are based on data present in the CATS Level 1B data products (v3.00). From the measurements featured in this product, we have used the Total Attenuated Backscatter (TAB) and the Perpendicular Attenuated Backscatter (PAB) at 1064 nm. From the same product, we have in addition used ancillary meteorological data,

https://doi.org/10.5194/egusphere-2025-5018 Preprint. Discussion started: 14 November 2025

© Author(s) 2025. CC BY 4.0 License.

including profiles of molecular backscatter at 1064 nm, of pressure and temperature. These come from the MERRA-2 reanalysis (Gelaro et al., 2017), extracted along the CATS surface footprint, and were provided on their own spatial grid. For the present paper we interpolated these profiles on the vertical and horizontal grid used for CATS backscatter measurements. To obtain the local time of observation for a CATS profile, its UTC time was offset considering the measurement longitude at the time of observation.

# 2.2 Ice crystal shape classification

## 1.1.1 Subsubsection (as Heading 3)

We performed the crystal shape classification by following the same steps as in Sato and Okamoto (2023), with small adaptations to the CATS configuration which are described hereafter.

First, we averaged all the CATS vertical profiles of TAB on the CloudSat-inspired spatial grid designed for the KU cloud product – 1 km horizontally, and 240 m vertically (Cesana et al., 2016). This averaging configuration provided appropriate signal quality for shape classification on CALIPSO data. Although CATS and CALIPSO operated at different wavelengths, their signal to noise ratios (SNR) at the same spatial resolutions are similar, which leads to very similar ice cloud detections in the Tropics (Sellitto et al., 2020). Thus, we suppose the KU averaging scheme designed for CALIPSO, which provides a good basis for ice crystal shape classification, will be equally appropriate when applied on CATS data. CATS L1B profiles are distant of ~350 m horizontally and contain points distant of 60 m vertically, thus we averaged 3 profiles horizontally and 4 points vertically.

Second, we applied to the averaged TAB profiles the cloud detection methodology described in Hagihara et al. (2010). This two-part detection scheme first identifies when the tropospheric TAB gets larger than a threshold including the backscattered signal from molecules, a possible contribution by aerosols at low altitudes, and instrumental noise derived from close measurements made in an elevated, supposedly clear-sky area (19-20 km above sea level or ASL). The Hagihara et al. (2010) detection method was designed for CALIPSO measurements at 532 nm, whereas most CATS data were observed at 1064 nm. At 1064nm the observed molecular backscattering signal is much weaker, which means its contribution to the threshold detection level will be mostly negligible. Moreover, since we are here only considering high-level clouds, the contribution of aerosols to the measured backscatter can be considered negligible, and our threshold was derived solely from instrumental noise. In a second pass, false detections were cleared up by a spatial consistency test considering a 9x9 bins window (9 km horizontally x 2.16 km vertically). This led to a spatially consistent cloud mask with a limited number of false detections.

Third, we identified ice clouds based on temperature (colder than  $-5^{\circ}$ C) and the x parameter. This parameter quantifies within a given profile the change in ATB between two consecutive 240m altitude levels inside a cloud layer, and can be considered a simple proxy for cloud extinction. Ice clouds, whose extinction is significantly weaker than liquid clouds, can

https://doi.org/10.5194/egusphere-2025-5018 Preprint. Discussion started: 14 November 2025

© Author(s) 2025. CC BY 4.0 License.

be identified by x 

Figure 1: Variation of crystal shape repartition with latitude for a) low-level clouds (pressure larger than 680 hPa), b) mid-level clouds (pressure between 440 and 680 hPa), and high clouds (pressure weaker than 440 hPa) according to CATS data between March 2015 and October 2017. Months between November and March are less present in the data (see text). The sum of frequencies for all shapes at a given pressure level and latitude is unity. d) total count of cloud points in the three pressure ranges according to latitude.

The amount of detected low-level (blue in Fig. 1d) and mid-level (orange) clouds are not symmetrical, instead both cloud types are more numerous in the North Hemisphere, most likely in relation with the continental hemispherical imbalance. Due to the slight under-representation of the north hemisphere winter November-February period (Sect. 2.1) in the sampled dataset, the position of the Inter-Tropical Convergence Zone (ITCZ) will be biased to the North, which could have an impact on results. Deliberately ignoring retrievals between March and September 2016 (thus uniformizing the seasonal sampling) has however no impact on the asymmetry of low and mid-level cloud amounts (not shown), or on any of the following results. High-level clouds (green) are the most numerous and follow a symmetrical trimodal zonal distribution, with high cloud amounts near the equator (inside the ITCZ), but also at midlatitudes.








In low-level clouds, the distribution of particle types (Fig. 1a) appears strictly symmetric around the equator. Below ~20° of latitude, ice particles in those clouds are almost exclusively liquid (blue). Solid particles appear near ~20° and become more frequent at higher latitudes. Among those particles, most frequent shapes are bullets (red), then 2D columns (green). These results are very consistent with CALIPSO-based ones in Sato and Okamoto (2023), as is the result that 2D plates, droxtals and Voronois are a minority. However, while those appear in roughly equal measures here, results based on CALIPSO data report almost no 2D plates and relatively more Voronoi shapes. We tried to consider only nighttime CATS data obtained at the same local time as CALIPSO (~1 AM, Fig. A1 in Appendix) to ensure we sampled the same part of the diurnal cycle; this did not change the results significantly. This suggests that CALIPSO retrievals provide a good representation of diurnally-averaged cloud properties. The differences found here are most likely not related to any diurnal cycle effect, but possibly to larger noise in the CATS depolarization signal compared to CALIPSO.

Considering mid-level clouds (Fig. 1b), liquid particles still follow a symmetric repartition and dominate the 20°S-20°N region. Their maximum frequency near the equator reaches 0.45. At higher latitudes, the importance of liquid particles drops to ~0.15 near 40° and ~0.1 at 55°. All these results are very consistent with CALIPSO ones. By comparison, the distributions of solid particles show a slight hemispheric asymmetry, as their maximum and/or minimum frequencies are not centered on the equator. Apart from liquid, the most frequent particle type is 3D bullets (red), and they dominate at latitudes higher than ~30°. Within the Tropics, they are clearly less present in the southern hemisphere, reaching their minimum frequency near ~15°S. The next most frequent particle type is 2D column (green), with a frequency that remains relatively stable zonally (in the 0.15-0.25 range) and a slight maximum in the southern hemisphere (near ~30°S). The three most frequent particle types (liquid, 3D bullets, 2D columns) are the same as those found in low-level clouds (Fig. 1c). These results are all very consistent with CALIPSO ones. The remaining particle types have comparable and small frequency ranges in CATS results (0.05-0.12) but their latitude distributions are different: 2D plates reach their maximum frequency in the southern hemisphere near ~20°S, where the least droxtals and Voronois are found. CALIPSO results suggest a clear dominance of Voronois and Droxtals, and a minority of 2D plates, especially at higher latitudes.

Finally, in high-level clouds (Fig. 1c) liquid particles (blue) are almost non-existent, which is not surprising. A slight increase in liquid particles is found near both 30°S and 30°N, a feature also found in CALIPSO-based results (Sato and Okamoto, 2023). This increase could be related to the local minimum of ice cloud amount in the extratropics (green, Fig. 1d), which would lead to an increased influence of detection sensitivity. Considering now solid particles, their amount are all symmetric around the equator, in agreement with CALIPSO results. CATS results report that Voronois (brown) dominate near the equator, with frequencies near 0.35 in the 20°S-20°N range, and that their importance drops as latitude increases. At latitudes higher than 20°, 3D bullets (red) become the dominant particle type with frequencies rising to 0.4 near 55° latitudes. This is in agreement with previous studies (e.g. Chepfer et al., 2001) which reported that polycrystals and hexagonal columns occur most frequently globally. While Sato and Okamoto (2023) agree on the importance of Voronois and 3D bullets based on CALIPSO data, and on how their frequencies evolve with latitude, they report that bullet dominate at all latitudes, even near the equator. Therefore, compared to CALIPSO results, CATS-based results report fewer bullets near the





equator. The other particle types show almost no latitude dependence in their repartition. CATS and CALIPSO results agree very well on the importance and latitude dependence of the remaining particle types: droxtals (purple) frequencies are near 0.2 around the equator and drop slightly at higher latitudes, 2D columns (green) are near 0.1 around the equator and increase slightly with latitude, and finally 2D plates are rare everywhere (less than 0.05).

To sum up, the CATS results shown here agree very well with the CALIPSO results reported in Sato and Okamoto (2023):

- in low and mid-level clouds, apart from liquid particles which dominate the Tropics, the most frequent particle types are 3D bullets and 2D columns;
- high clouds feature mostly 3D bullets and Voronois (the second one being significantly more frequent in the 20°S-20°N band) followed by droxtals, and finally the other types far behind with little latitude dependence;
- the distribution of particle types is roughly symmetric around the equator, except mid-level clouds which feature significant asymmetries.

These results appear independent of CATS's seasonal or diurnal sampling. In the rest of the paper, the CATS-based results we present have no CALIPSO equivalent that we are aware of at this time.

## 175 **3.2** Particle type partitioning with temperature

In this section we document how particles are distributed among categories depending on the atmospheric temperature, in three zonal groups: South hemisphere midlatitudes (30°S-50°S), Tropics (30°S-30°N), and North Hemisphere midlatitudes (30°N-50°N). We considered separately results obtained with and without incoming sunlight (hereafter called daytime and nighttime results), according to the CATS granule classification.

In nighttime conditions (18:00-06:00 local time, Fig. 2), the partitioning of cloud particle categories according to temperature is very similar in the three zonal regions, and almost exactly the same at midlatitudes in both hemispheres. As expected, the fraction of liquid particles (blue) falls from 20% near -10°C to 0 near -40°C. The fraction of 2D plates (orange) and columns (green) also decreases with colder temperatures. Although 2D plates remain generally negligible at all temperatures, 2D columns are quite frequent at warm temperatures (~20% fractions) and remain noticeable even at -80°C. The importance of 2D columns drops faster in the Tropics (Fig. 2b) compared to midlatitudes (Fig. 2a and 2c). 3D bullets (red) are present at all temperatures, and dominate the repartition at midlevel temperatures (-30°C to -50°C). At temperatures below -50°C, they are more frequent in midlatitudes than in the Tropics. Droxtals (purple) and Voronois (brown) become more frequent as temperatures get colder. Together, they make for more than half of particles at temperatures below -60°C in midlatitudes, and below -50°C in the Tropics.




Figure 2: Partitioning of cloud particle categories in a) south hemisphere midlatitudes, b) tropics, and c) north hemisphere midlatitudes according to temperature. At each temperature the sum of all fractions is unity. d) total count of cloud points in the three latitude bands according to temperature.

In daytime conditions (06:00-18:00 local time, not shown), results are very similar. Notable differences include the fraction of 2D plates being generally larger (never exceeding 10% though) and constant at all temperatures. In the Tropics, the partitioning of cloud particles appears very stable at temperatures colder than -40°C, with only a limited growth of Voronois at the expense of 3D bullets. This behavior is not found at midlatitudes, where results are consistent with those found in nighttime conditions. Day-night differences might be due to noisier depolarization ratio in the CATS daytime data.

# 3.3 Particle type partitioning with local time

So far, we have presented the variation of the relative fraction of each particle type according to latitude (Sect. 3.1) and temperature (Sect. 3.2). Here we investigate how, in each latitude and temperature bin, this relative fraction changes along a day. From the relative fraction at each hour, we subtracted the daily average from each hourly fraction to compute the diurnal fraction anomaly (DFA) and document its cycle during the day. Fig. 3 shows the DFA cycle in the Tropics (30°S-30°N) for each temperature range and particle type (DFA cycles in the North and South midlatitudes are shown in Fig. A2 and A3 in the Appendix). In the majority of cases, DFAs feature at least one noticeable maximum, occurring either in daytime (8:00-16:00) or nighttime (20:00-4:00), and a noticeable minimum at the opposite point of the diurnal cycle. The difference between the minimum and the maximum of the DFA is its amplitude. The amplitude in general gets larger at colder temperatures, and flatter at warmer temperatures. In the rest of the section, we focus on this amplitude as an indicator of the diurnal fluctuations of the importance of a given particle shape.



Tropics (30S-30N)

Figure 3: Diurnal variation of the fraction anomaly in the Tropics (30°S-30°N) for each particle type (columns) and temperature range (rows). The subplot background color indicates whether the diurnal variation has a maximum during daytime (red) or nighttime (blue). The intensity of the color provides an indication of the nighttime vs daytime diurnal amplitude. The variations shown here are relative to the daily averages of fractions, which are shown in Fig. 2 according to temperature (Sect. 3.2). As a reminder, for each temperature range and particle type, the daily average fraction is indicated in the top left (the sum of daily fractions is the unity in a given temperature range).

Figure 4 documents the amplitude of DFAs as a function of temperature in the three zonal regions considered so far. The DFA amplitude and the average fraction are not independent: large daily variations of fractions are only possible when the average fraction is important. For instance, at temperatures colder than -40°C liquid particles are inexistent – as a consequence, at such temperatures the DFA amplitude is zero.



Figure 4. Amplitude of the diurnal fraction anomaly (DFA) for each particle type, as a function of temperature, in a) the South hemisphere midlatitudes, b) the Tropics, and c) the North hemisphere midlatitudes (right). The vertical axis is the difference between the minimum and maximum of daily cycles shown in Fig. 3. In areas shaded red, the daily maximum is reached during daytime. In areas shaded blue, the daily maximum is reached during nighttime. For instance, in South midlatitudes (Fig. 4a), at -80°C, particle types that follow a strong diurnal cycle are 2D plates, 2D columns, and droxtals. These diurnal cycles make 2D plates and columns more frequent during the daytime, and droxtals during the nighttime.

The DFA of liquid (blue lines in Fig. 4), 2D plates (orange) and 2D columns (green) almost always feature a positive amplitude, meaning the fraction of these particles follows a daily cycle with a marked daytime maximum. For liquid particles, this amplitude gets larger from -40°C to 0°C, meaning the daytime maximum and nighttime minimum get more clearly separated. Near 0°C, liquid particles are the only ones that feature a significant daily cycle. These findings for liquid particles are true in the three zonal regions. 2D plates and 2D columns also follow a cycle with a daytime maximum, but its amplitude is largest at cold temperatures and decreases as temperatures get warmer. At the warmest temperatures (-20°C to 0°C), the DFA amplitude of 2D plates reaches 0, meaning the daily cycle gets flat. The DFA amplitude of 2D columns even gets slightly negative in midlatitudes (Fig. 3a and 3c): at warm temperatures their DFA maximum happens during nighttime.






The DFA of Droxtals (purple) always shows a negative amplitude, which gets closer to 0 at warmer temperatures. This means these particles reach their maximum presence during nighttime, with a strongest diurnal cycle at coldest temperatures. The DFA of 3D bullets (red) also mostly shows a negative amplitude. In midlatitudes, this amplitude reaches furthest away from 0 in the -60°C to -40°C temperature range, and gets quite weak at warmest and coldest temperatures, especially in the South hemisphere (Fig. 3a) where the daily cycle completely disappears. By contrast, in the Tropics, at very cold temperatures, the amplitude gets markedly positive below -60°C, meaning that in this temperature range the daily cycle reaches its maximum during daytime. 3D bullets are the only particle type that features a transition between a strong daytime maximum (e.g. in the Tropics near -80°C) and a strong nighttime maximum (e.g. in the Tropics near -30°C).

Finally, the DFA amplitude of Voronois (brown) is generally weak, except when colder than -40°C in the Tropics. In such conditions, Voronoi particles follow the strongest daily cycle of all particle types, with a strongly marked nighttime maximum. A feature not captured by Fig. 4 is the development for Voronoi particles in the Tropics at temperatures warmer than -40°C of a more complex cycle (Fig. 3, right column), featuring over a daily period two maxima near 6:00 and 18:00, and two minima near noon and midnight. The amplitude of this unusual, 12-hour cycle is quite strong in the Tropics at midlevel temperatures, but gets weaker at warmer temperatures and is almost unnoticeable in midlatitudes (Fig. A2 and A3).

# 250 4. Conclusions

It has long been hinted that measurements of depolarization ratio contain qualitative information on the shape of particles being probed (e.g. Noel et al., 2004; Midzak et al., 2020). Okamoto et al. (2019) developed a framework to partition ice cloud particles into 6 categories with different optical signatures and microphysical features, which Sato and Okamoto (2023) applied to CALIPSO observations. Here we applied the framework on 33 months of data from the CATS spaceborne lidar. Statistics on the results have enabled documenting the evolution of this partitioning as a function of local time, latitude and temperature. After confirming that results from CATS were consistent with those from CALIPSO (Sect. 3.1), we showed how particle types in ice cloud transition from a dominance of simple liquid, plate-like and columnar particles at warm temperatures (-20°C and up) to a dominance of more complex shapes (Voronois and droxtals) at coldest temperatures (-60°C to -80°C, Sect. 3.2). This temperature-dependent transition, which has been long suggested by global statistics of depolarization ratio (as in Sassen et al., 2012), appears zonally stable, meaning existing differences in cloud formation processes between the Tropics and midlatitudes only marginally impact the relationship between particle type and temperature. Finally, we documented the daily cycle of the partitioning between particle types in different temperature ranges and zonal regions (Sect. 3.3). Particles with strong daily cycles include 2D columns and plates at cold temperatures (daytime maximum), 3D bullets and droxtals at warm temperatures (nighttime maximum), and Voronois at very cold temperatures (nighttime maximum). 3D bullets are the only particles found to transition from a daytime maximum at coldest temperatures to a nighttime maximum at warmest temperatures. As far as we can tell, these results are seasonally stable.





Limitations to this study include that the ice cloud averaging and detection process prevents the inclusion of clouds vertically thinner than 2.16 km and horizontally smaller than 9 km. Thus, our results do not apply to e.g., very thin cirrus clouds near the tropopause, which are frequently vertically thinner (Martins et al., 2011; Lesigne et al., 2024) and could present unusual microphysical properties due to their formation process which can involve small-scale atmospheric dynamics in subsaturated regions (Kärcher et al., 2024). Future work should include developing classification methodologies specifically suited to these clouds. Although the results of the present classification appear reasonable and consistent with our understanding of the geographical variability of ice crystal shapes, conclusions would be validated by finding ways to confirm results of the classification process. Future work, therefore, involves confronting such results with other information on particle microphysical properties, for instance from in-situ retrievals from airborne or balloon probes. Moreover, even when considering relatively thick ice clouds, it is not unreasonable to suppose that the type of particles within will depend on the processes leading to cloud formation, either convective or stratiform (Reverdy et al., 2012). Future work thus also involves confronting the retrieved particle types with macrophysical or morphological properties of clouds, for instance derived from collocated sun-synchronous satellite instruments (Bouniol et al., 2021). In any case, a primary concern should be to improve consistency between categories of particles based on distinct optical signatures and particle types being implemented in climate and mesoscale atmospheric models, so retrievals from measurements provide useful constraints to simulations (Xu et al., 2023). Finally, future work involves adapting the method presented here to measurements from other spaceborne lidar missions such as EarthCARE (Wehr et al., 2023), or from upcoming missions, such as Luce or TOMCAT (Yorks et al., 2023). Combining retrievals from consecutive spaceborne lidar missions, by enabling long-term datasets of microphysical properties of ice clouds, could help identify changes related to anthropogenic climate change (Chepfer et al., 2018).

#### Data availability

The CATS Level 1B products (doi: 10.5067/ISS/CATS/L1B\_N-M7.2-V3-00) are distributed by ASDC: <a href="https://asdc.larc.nasa.gov/project/CATS-ISS/CATS-ISS\_L1B\_N-M7.2-V3-00\_V3-00">https://asdc.larc.nasa.gov/project/CATS-ISS/CATS-ISS\_L1B\_N-M7.2-V3-00\_V3-00</a>. For the current study they were analyzed through the AERIS/ICARE service <a href="https://www.aeris-data.fr/icare/">https://www.aeris-data.fr/icare/</a>

# 290 Author contributions

VN designed the study, created the figures and wrote the first draft. HC provided guidance on the use of spaceborne lidar datasets. CB provided expertise on cloud microphysics and atmospheric models. JY provided expertise on the analysis of CATS data. HC, CB and JY provided feedback on the manuscript.

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

# Appendix

Figure A1. Same as Fig. 1, but considering only clouds detected at local times between midnight and 2 AM, to sample the same part of the diurnal cycle as CALIPSO.

## midlatN (30N-50N)

Figure A2. Like Fig. 3, in the North hemisphere midlatitudes (30°N-50°N).

## midlatS (30S-50S)

Figure A3. Like Fig. 3, in the South hemisphere midlatitudes (30°S-50°S).