# Peer review of "The diurnal cycle and temperature dependence of crystal shapes in ice clouds from satellite lidar polarized measurements"

_EGUsphere, 2025_

## Referee Comment (RC2)

Title: The diurnal cycle and temperature dependence of crystal shapes in ice clouds from satellite lidar polarized measurements
Author(s): Vincent Noel, Hélène Chepfer, Christelle Barthe, and John Yorks
MS No.: egusphere-2025-5018
MS type: Research article
Iteration: Initial submission

**General Comments:**

This paper uses the methodology of Sato and Okamoto (2023) that was applied to CALIPSO lidar (CALIOP) measurements to estimate the relative abundance of various ice particle shapes in clouds but now applies this methodology to the CATS (Cloud Aerosol Transport System) satellite lidar dataset for this same purpose. As a "sanity check", consistency between the results from this new study and that of Sato and Okamoto (2023) was verified. Then the diurnal variation of ice particle shape was investigated for the first time, with quite interesting results. This paper is of high caliber and worthy of publication in ACP after minor revision by addressing the comments listed below. The paper is well organized and well written.

**Specific Comments:**

1. Section 2.2: As shown by Eq. 1 in Sato and Okamoto (2023), the lidar backscatter $\beta$ is an integral product of the particle size distribution (PSD) and the particle's mean backscattering cross-section ($C_{bk}$) where $C_{bk}$ depends on particle size. Thus, $\beta$ appears to be a measure of the PSD second moment, while the ice particle number concentration $N_i$ denotes the $0^{th}$ moment of the PSD. Is it safe to relate the fraction of a particle shape in this paper to the relative $N_i$ of that particle shape in the clouds? That is, there may be a tendency for readers to interpret these results as a relative measure of $N_i$ for each ice particle shape. Since the lidar depolarization ratio $\delta$ that is used to discriminate cloud particle shape is the ratio of two $\beta$ values (for horizontal and vertical polarization), PSD effects should cancel, leaving just the depolarization effect. The statistics in this paper would thus be reporting the frequency of occurrence of $\delta$ corresponding to various cloud particle shape categories as defined in Sato and Okamoto (2023), where $\delta$ identifies the dominant shape sampled.

   While in essence this is implied in Sect. 2.2 (and is evident in Sato and Okamoto), more of this information could be presented so that the reader can more clearly understand what the statistics in this paper actually mean.

2. The paper would be more interesting if it included images for the different ice particle shapes being evaluated. Voronois ice particles are especially important since many readers may not be familiar with them, and several images may be justified due to their varied, complex shapes.

3. Lines 251-252: Please cite Ken Sassen's work from the 1990's here. Ken was the first to relate lidar depolarization ratios to cloud particle shape as per my understanding, and he has many published papers on this topic.

Technical Comments:

1. Line 69: This line contains "**1.1.1 Subsection (as Heading 3)**" and should be deleted.
2. Line 92: ATB => TAB?
3. Line 107: -80°S => -80°C ?